# Amoxicillin/clavulanate in combination with rifampicin/clarithromycin is bactericidal against *Mycobacterium ulcerans*

Emma Sáez-López[1,2]*, Ana C. Millán-Placer[1,2¤], Ainhoa Lucía[1,2], Santiago Ramón-García[1,2,3]*

**1** Department of Microbiology, Paediatrics, Radiology and Public Health, Faculty of Medicine, University of Zaragoza, Zaragoza, Spain, **2** Spanish Network for Research on Respiratory Diseases (CIBERES), Carlos III Health Institute, Madrid, Spain, **3** Research & Development Agency of Aragón (ARAID) Foundation, Zaragoza, Spain

¤ Current address: Operon S.A., Cuarte de Huerva, Zaragoza, Spain.
* esaez@unizar.es (ES-L); santiramon@unizar.es (SR-G)

## Abstract

**Data Availability Statement:** The authors confirm that all data underlying the findings are fully available without restriction. All relevant data are

### Background

Buruli ulcer (BU) is a skin neglected tropical disease (NTD) caused by *Mycobacterium ulcerans*. WHO-recommended treatment requires 8-weeks of daily rifampicin (RIF) and clarithromycin (CLA) with wound care. Treatment compliance may be challenging due to socioeconomic determinants. Previous minimum Inhibitory Concentration and checkerboard assays showed that amoxicillin/clavulanate (AMX/CLV) combined with RIF+CLA were synergistic against *M. ulcerans*. However, *in vitro* time kill assays (TKA) are a better approach to understand the antimicrobial activity of a drug over time. Colony forming units (CFU) enumeration is the *in vitro* reference method to measure bacterial load, although this is a time-consuming method due to the slow growth of *M. ulcerans*.

The aim of this study was to assess the *in vitro* activity of RIF, CLA and AMX/CLV combinations against *M. ulcerans* clinical isolates by TKA, while comparing four methodologies: CFU enumeration, luminescence by relative light unit (RLU) and optical density (at 600 nm) measurements, and 16S rRNA/IS*2404* genes quantification.

### Methodology/Principal findings

TKA of RIF, CLA and AMX/CLV alone and in combination were performed against different *M. ulcerans* clinical isolates. Bacterial loads were quantified with different methodologies after 1, 3, 7, 10, 14, 21 and 28 days of treatment.

RIF+AMX/CLV and the triple RIF+CLA+AMX/CLV combinations were bactericidal and more effective *in vitro* than the currently used RIF+CLA combination to treat BU. All methodologies except IS*2404* quantitative PCR provided similar results with a good correlation with CFU enumeration. Measuring luminescence (RLU) was the most cost-effective methodology to quantify *M. ulcerans* bacterial loads in *in vitro* TKA.

within the paper and its Supporting Information files.

**Funding:** This work was partially supported by grants from the Tres Cantos Open Lab Foundation (Grant No. TC281) and from Anesvad Foundation, both to SRG. ACMP was recipient of a fellowship from the Spanish Government (Programa de Formación de Profesorado Universitario) Ref. FPU 17/01812. The funders had no role in study design, data collection and analysis, decision to publish, or preparation of the manuscript.

**Competing interests:** The authors have declared that no competing interests exist.

## Conclusions/Significance

Our study suggests that alternative and faster TKA methodologies can be used in BU research instead of the cumbersome CFU quantification method. These results provide an *in vitro* microbiological support to of the BLMs4BU clinical trial (*NCT05169554, PACTR202209521256638*) to shorten BU treatment.

### Author summary

Since 2004, when only surgery was available, Buruli ulcer (BU) treatment has improved reaching to the efficient 8-weeks all oral antibiotic course of rifampicin and clarithromycin together with wound care and, sometimes, tissue grafting and surgery. This skin neglected tropical disease caused by *Mycobacterium ulcerans* mainly affects people living in rural areas in under-resourced countries with limited access to health services and medicines, thus compromising patients' treatment adherence. The inclusion of amoxicillin/clavulanate in BU therapy was previously described *in vitro* with the potential to shorten BU treatment.

In this study, we confirmed the high bactericidal activity over time of rifampicin and amoxicillin/clavulanate-containing combinations against *M. ulcerans*, being even more effective *in vitro* than the antibiotic combination currently used to treat BU. A comparison of different methodologies that are applied in the laboratory and in the clinic showed good correlation between them, leading to a wide variety of biomarkers for BU research and giving the opportunity for clinical translation; the choice of the most suitable one being driven by the purpose and the context of the study. These *in vitro* data provide further support to the ongoing clinical trial in West Africa to evaluate if BU treatment can be shortened from 8 to 4 weeks (BLMs4BU trial: *NCT05169554, PACTR202209521256638*).

## Introduction

Buruli ulcer (BU) is a skin neglected tropical disease (NTD) caused by *Mycobacterium ulcerans* that affects skin, soft tissues and bones. Its mode of transmission remains unknown, making prevention strategies a challenge. Therefore, early diagnosis and treatment are crucial to minimize morbidity, stigma, costs associated with the disease and to prevent long-term disability [1]. WHO-recommended treatment requires 8-weeks of daily rifampicin (RIF) and clarithromycin (CLA) with extensive wound care, and sometimes tissue grafting and surgery [1]. BU generally affects poor communities in remote rural areas with limited access to health services. Difficulties in the access to medicines together with the need of prolonged hospitalization (healing can take up to one year) impacts household's income and can lead to a low adherence to the treatment by the patients [2].

Arenaz *et al.* [3] described the potential inclusion of amoxicillin/clavulanate (AMX/CLV) in BU therapy as a prospect for shortening the length of treatment and time to cure. Minimum Inhibitory Concentration (MIC) and checkerboard assays showed that betalactams combined with RIF and CLA were synergistic against *M. ulcerans*; these assays were based on growth inhibition readouts at a fixed time point [4–6]. *In vitro* time-kill assays (TKA) quantify the antibiotic concentration-effect relationship in a time-dependent manner and provide a

measure of cidality, instead of growth inhibition, with more granular information on the degree of drug interactions.

Quantification of colony forming units (CFU) is the *in vitro* reference method to measure bacterial load of a culture. In the case of pathogenic slow growing mycobacteria, this is a burdensome method (including the need to work in Biosafety Level (BSL) 3 laboratories) because colony formation can take up to three months, as is the case of *M. ulcerans*. Alternative methodologies are available nowadays for assessing various viability parameters, such as cell proliferation, cell membrane integrity and active metabolism [7]. For instance, a relationship between optical density measurements at a wavelength of 600 nm ($OD_{600}$) and CFU quantification has been reported in mycobacteria [8]. Similarly, the luminescence relative light unit (RLU) assay and CFU counting positively correlated in *in vitro* and *in vivo* studies using bioluminescent *M. ulcerans* strains [9–13]. Moreover, Beissner *et al.* [14] developed a combined 16S rRNA reverse transcriptase real-time PCR/IS*2404* quantitative real-time PCR (qPCR) assay, which proved to be highly sensitive, specific and efficient in detecting viable *M. ulcerans* in clinical samples under field conditions [15]. Nonetheless, there is a lack of appraisal on which methodology would be the most appropriate since, to date, no study has evaluated all four readouts in parallel.

The aim of this study was thus to assess the *in vitro* activity of RIF, CLA and AMX/CLV combinations against clinical *M. ulcerans* isolates by TKA, while comparing the four above-mentioned readouts: CFU enumeration, luminescence RLU and $OD_{600}$ measurements, and quantification of 16S rRNA/IS*2404* genes by qPCR.

## Material and methods

### Bacterial strains, general growth conditions and reagents

*M. ulcerans* NCTC 10417 (ATCC number: 19423) and clinical isolates from different geographical origins: ITM 000932, ITM 941327 and ITM C05142, Australia; ITM 063846, Benin; ITM 070290, China; ITM C05143, Mexico; and ITM C08756, Japan, were purchased from the Belgian Coordinated Collection of Microorganisms (BCCM). *M. ulcerans* cells were grown at 30˚C in Middlebrook 7H9 broth supplemented with 0.2% glycerol, 0.05% (vol/vol) Tyloxapol and 10% oleic acid-albumin-dextrose-catalase (OADC). Rifampicin (R3501-250MG), clarithromycin (C9472-100MG), amoxicillin (A8523-1G) and clavulanate (Potassium clavulanate, 33454-100MG) were purchased from Sigma.

### Bacteriologic assessment by Colony Forming Units (CFU) enumeration

A total of 100 μL of the bacterial cultures, including seven 10-fold serial dilutions, were plated in duplicates onto Middlebrook 7H10 agar media plates supplemented with 10% OADC and incubated at 30˚C up to three months, when CFU per unit volume of culture (mL) were visually enumerated. Bacterial load was reported as $Log_{10}$ CFU/mL.

### Determination of the optical density

A volume of 600 μL was pipetted from the cultures into a cuvette and a spectrophotometer (Biochrom Colourwave WPA CO7500) was used to measure the absorbance at a wavelength of 600 nm ($OD_{600}$). The background absorbance of the 7H9 media was subtracted from the signal.

### BacTiter-Glo Microbial Cell Viability Assay

A luminescence method (BacTiter-Glo Microbial Cell Viability Assay Kit, Promega [16]) based on ATP quantitation was used to determine the number of viable bacterial cells in broth

culture. With this methodology, the amount of light produced is dependent on the amount of ATP in the sample. Fifty microliters of BacTiter-Glo Reagent were added to an equal volume of bacterial culture in an opaque walled 96-well plate in duplicates. Plates were incubated for five minutes at room temperature before luminescence (Relative Light Units, RLU) was recorded in a luminometer (BioTek Synergy HTX Reader—Multimode Microplate Readers).

## Combined 16S rRNA Reverse Transcriptase/IS*2404* Real-Time qPCR assay

One millilitre of each culture was centrifuged for 5 minutes at 10,000 rpm. The pellet was resuspended in a wash buffer and stabilized by RNA protect (Qiagen, Hilden, Germany). Simultaneous extraction of RNA/DNA was performed by the AllPrep DNA/RNA Mini extraction kit (Qiagen) as previously described by Beissner *et al*. [14], with slight modifications. Here, mechanical lysis was performed by FastPrep (6.5 m/s, 45 s; twice) after the stabilisation of the cultures. DNA extracts (100 μL) were stored at -20˚C, whereas RNA extracts (50 μL) were stored at -80˚C. To remove potential genomic DNA (gDNA) from the *M. ulcerans* extracted RNA, the TURBO DNase (TURBO DNA-free kit, Invitrogen) was used according to manufacturer instructions. Before transcription, a 2 μL aliquot of *M. ulcerans* RNA (gDNA free) was used to perform the IS*2404* qPCR assay for RNA quality control. From the remaining *M. ulcerans* RNA extract, 12 μL were reverse transcribed into cDNA using the QuantiTect Reverse Transcription Kit (Qiagen), as described elsewhere [14]. DNA and cDNA were subjected to IS*2404* qPCR and 16S rRNA RT-qPCR, respectively [14]. All samples were performed in triplicate and each run included negative controls ("no template"). Standard curves were performed by including eight 10-fold serial dilutions from a plasmid with $10^8$ gene copies/μL of IS*2404* (GenExpress, 30-8606-01) and 16S rRNA (GenExpress, 30-7522-01).

## *M. ulcerans* time-kill assays

TKA were performed to measure the bacterial load of the different *M. ulcerans* clinical isolates upon treatment with RIF, CLA and AMX/CLV both alone and in pairwise and triple combinations. Briefly, an exponential growth phase ($OD_{600}$ = 0.5–1) culture was diluted to approximately $10^4$–$10^5$ CFU/mL (Day -3) and allowed to grow for three days. Then, drugs were added (Day 0) and the bacterial load was quantified after 0, 1, 3, 7, 10, 14, 21 and 28 days of treatment initiation. To cover the range of MIC values across the clinical isolates (S1 Table) [3] the same drug concentrations were tested against all the isolates: RIF, 0.1 μg/mL; CLA, 0.25 μg/mL; and AMX/CLV, 0.5 μg/mL. These concentrations were considered as the 1xMIC values and used for fold-MIC changes calculations. These were calculated by division of the MIC used for the experiments and the actual MIC for each isolate (S1 Table). For example, for RIF at 1/20xMIC ($MIC_{RIF}$ used in this study is 0.1 μg/mL), the antibiotic concentration used was 0.005 μg/mL. For the ITM 063846, its actual $MIC_{RIF}$ is 0.025 μg/mL. When 0.005 was divided by 0.025, the result showed the fold decrease (1/5 = 0.02 fold). When needed, CLV was added at a fixed concentration of 5 μg/mL.

Time-kill kinetics of isolate ITM 000932 were performed using bacterial counts (CFU/mL), luminescence (RLU), $OD_{600}$, and number of gene (16S rRNA and IS*2404*) copies using drugs at 1/4xMIC values (RIF, 0.025 μg/mL; CLA, 0.0625 μg/mL; AMX/CLV, 0.125 μg/mL). Time-kill kinetics of isolates ITM 941327, ITM C05142, ITM 063846, ITM 070290, ITM C05143, ITM C08756, ITM 000932, and NCTC 10417 were performed using luminescence (RLU) with drugs alone and in combination at 1/4xMIC (RIF, 0.025 μg/mL; CLA, 0.0625 μg/mL; AMX/CLV, 0.125 μg/mL) and 1xMIC values (RIF, 0.1 μg/mL; CLA, 0.25 μg/mL; AMX/CLV, 0.5 μg/mL).

## Statistical analysis

Raw data was analysed using the GraphPad Prism version 8.0 (GraphPad Software). $Log_{10}$ CFU/mL reduction across different conditions during the time treatment intervals were analysed using a model of analysis of variance (ANOVA) in one-way with post-test Tukey correction for multiple comparisons. One-way ANOVA was also used for the mean $OD_{600}$ values in the tested conditions. The Pearson or Spearman (for non-normal distribution) correlation coefficients were calculated to compare methodologies. A p-value below 0.05 was considered statistically significant.

## Results

### The luminescence readout is the most cost-effective methodology to quantify *M. ulcerans* bacterial loads in *in vitro* time-kill assays

Four methodologies were explored to assess *M. ulcerans* cell viability upon treatment with RIF, CLA and AMX/CLV alone and in combination; the gold standard CFU enumeration plus three additional methodologies: RLU and $OD_{600}$ measurements, and quantification of 16S rRNA/IS*2404* copy numbers. (**Fig 1**).

 Changes in the $Log_{10}$ CFU/mL of viable colonies showed that none of the compounds alone, neither RIF+CLA and CLA+AMX/CLV combinations, were able to reduce the bacterial burden more than one $Log_{10}$ CFU/mL at the end of the treatment period (Day 28) (**S2 Table**). However, RIF+AMX/CLV and RIF+CLA+AMX/CLV combinations exhibited a significant (P-value < 0.05) bactericidal activity against *M. ulcerans*, being the most active combinations with a bacterial load reduction at Day 28 of almost 3 $Log_{10}$ CFU/mL (>99,9% killing activity)

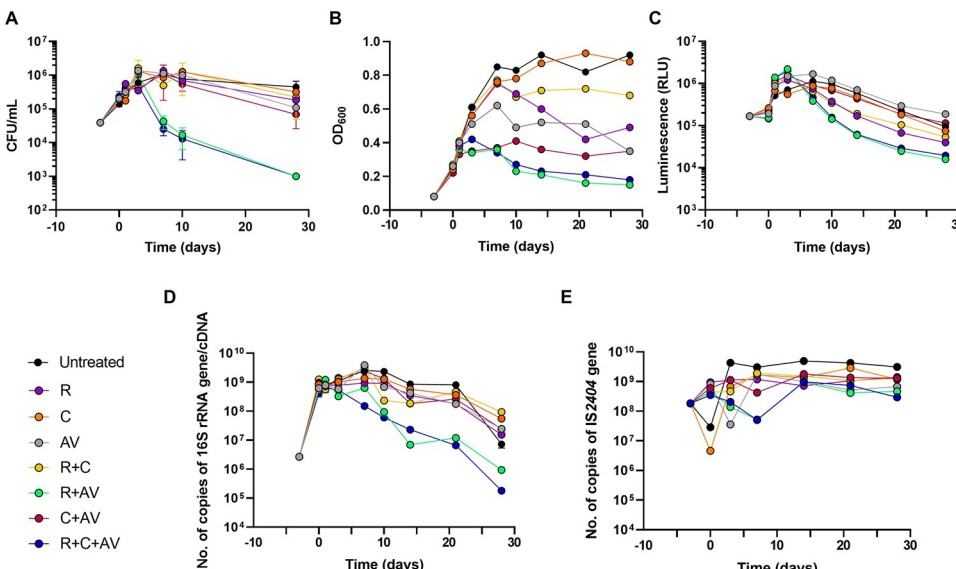

**Fig 1. Time-kill kinetics of rifampicin, clarithromycin and amoxicillin/clavulanate alone and in combination against *M. ulcerans* isolate ITM 000932 using different bacterial load quantification methodologies.** Bacterial load quantification by: (**A**) Colony Forming Units (CFU) enumeration; (**B**) Luminescence (Relative Light Units, RLU); (**C**) Culture optical density measurements at a wavelength of 600 nm ($OD_{600}$); (**D**) Number of copies of 16S rRNA gene/cDNA; and (**E**) Number of copies of the IS*2404* gene. Rifampicin, clarithromycin and amoxicillin were added at 1/4xMIC. Clavulanate was used at fixed 5 μg/mL. $MIC_{RIF}$ = 0.1 μg/mL; $MIC_{CLA}$ = 0.25 μg/mL; $MIC_{AMX/CLV}$ = 0.5 μg/mL. Values represent the mean of two or three measures from different experiments; error bars indicate the standard deviation. CFU, Colony Forming Units; RLU, Relative Light Units; OD, Optical Density; R, rifampicin; C, clarithromycin; AV, amoxicillin/clavulanate.

**Table 1. Correlation among methodologies used in this study to evaluate the activity of rifampicin, clarithromycin and amoxicillin/clavulanate alone and in combination against *M. ulcerans* isolate ITM 000932.** The Pearson correlation coefficient was calculated among the treatment groups.

| Culture condition | CFU *vs.* RLU | CFU *vs.* OD | CFU *vs.* Number of copies of 16S rRNA gene/cDNA | RLU *vs.* OD | RLU *vs.* Number of copies of 16S rRNA gene/cDNA | OD *vs.* Number of copies of 16S rRNA gene/cDNA |
|---|---|---|---|---|---|---|
| Untreated | 0.879[b] | 0.828[b] | 0.830[b] | 0.381 | 0.926[b] | 0.372 |
| RIF | 0.427 | 0.788[b] | 0.680[b] | 0.331 | 0.692[b] | 0.515 |
| CLA | 0.634 | 0.591 | 0.737 | 0.148 | 0.882[b] | 0.151 |
| AMX/CLV | 0.941[b] | 0.838[b] | 0.607[a] | 0.720 | 0.800[a,b] | 0.435[a] |
| RIF+CLA | 0.588[a] | 0.533 | 0.571[a] | (-)0.1500[a] | 0.733[a,b] | 0.233[a] |
| RIF+AMX/CLV | 0.893[a,b] | 0.685[a] | 0.714[a] | 0.695[a,b] | 0.700[a,b] | 0.946[a,b] |
| CLA+AMX/CLV | 0.531 | 0.811[a,b] | 0.929[b] | 0.435[a] | 0.611 | 0.519[a] |
| RIF+CLA +AMX/CLV | 0.893[a,b] | 0.607[a] | 0.857[a,b] | 0.800[a,b] | 0.867[b] | 0.917[a,b] |

[a]Spearman correlation coefficient was calculated

[b]$p < 0.05$. CFU, Colony Forming Units; OD, Optical Density; RLU, Relative Light Units; RIF, rifampicin; CLA, clarithromycin; AMX/CLV, amoxicillin/clavulanate.

(**Fig 1A** and **S2 Table**). Similar conclusions were extracted from luminescence (RLU) measurements (**Fig 1B**); indeed, the correlation coefficient between these two methodologies was 0.893 in the RIF+AMX/CLV and RIF+CLA+AMX/CLV combinations (**Table 1**). $OD_{600}$ measurements provided more scattered data but similar trends as with CFU enumeration and RLU quantification. Here, besides the RIF+AMX/CLV and RIF+CLA+AMX/CLV combinations, whose values did not surpass an absorbance of 0.42, the CLA+AMX/CLV combination also showed a mean significantly different (P-value <0.05) compared to the other conditions (**Fig 1C**). Finally, the decline in the number of 16S rRNA gene copies also confirmed RIF +AMX/CLV and RIF+CLA+AMX/CLV as the combinations with the strongest bactericidal interaction (**Fig 1D**). On the contrary, IS*2404* copy numbers remained stable in all conditions, which confirmed that no differentiation could be made between viable and nonviable bacteria with this methodology (**Fig 1E**). Therefore, correlation factors for IS*2404* qPCR were not included in **Table 1**.

In summary, an overall good correlation was observed between CFU enumeration and the other methodologies (**Table 1**). All strong relationships (r > ~0.700) were statistically significant (P-value < 0.05). The highest correlation coefficients were observed between RLU and the number of copies of 16S rRNA gene, both being metabolism activity indicators. Based on these results and from a technical practicality perspective, the luminescence (RLU) method was selected for further studies.

### The combination of rifampicin/clarithromycin plus amoxicillin/clavulanate is bactericidal against *M. ulcerans* clinical isolates

The RLU assay was used to evaluate the kill kinetics of the RIF, CLA, and AMX/CLV combinations against several *M. ulcerans* clinical isolates. The activity of the single compounds alone was first tested in dose response time-kill kinetic studies (**S1 Fig**). For most isolates, antibiotic concentrations of, at least, four times the MIC used in the experiments (4xMIC: RIF, 0.4 μg/mL; CLA, 1 μg/mL; and AMX/CLV, 2 μg/mL) were needed to observe a bactericidal effect of RIF and CLA alone, which correlated with an increase range of 2-16-fold of their actual MIC. AMX/CLV was less potent although a trend toward bactericidal activity was also observed at the highest concentrations. Based on these results, combination studies were performed at fixed 1/4xMIC and 1xMIC values in order to compare the efficacy of the combinations across all the isolates (**Fig 2**).

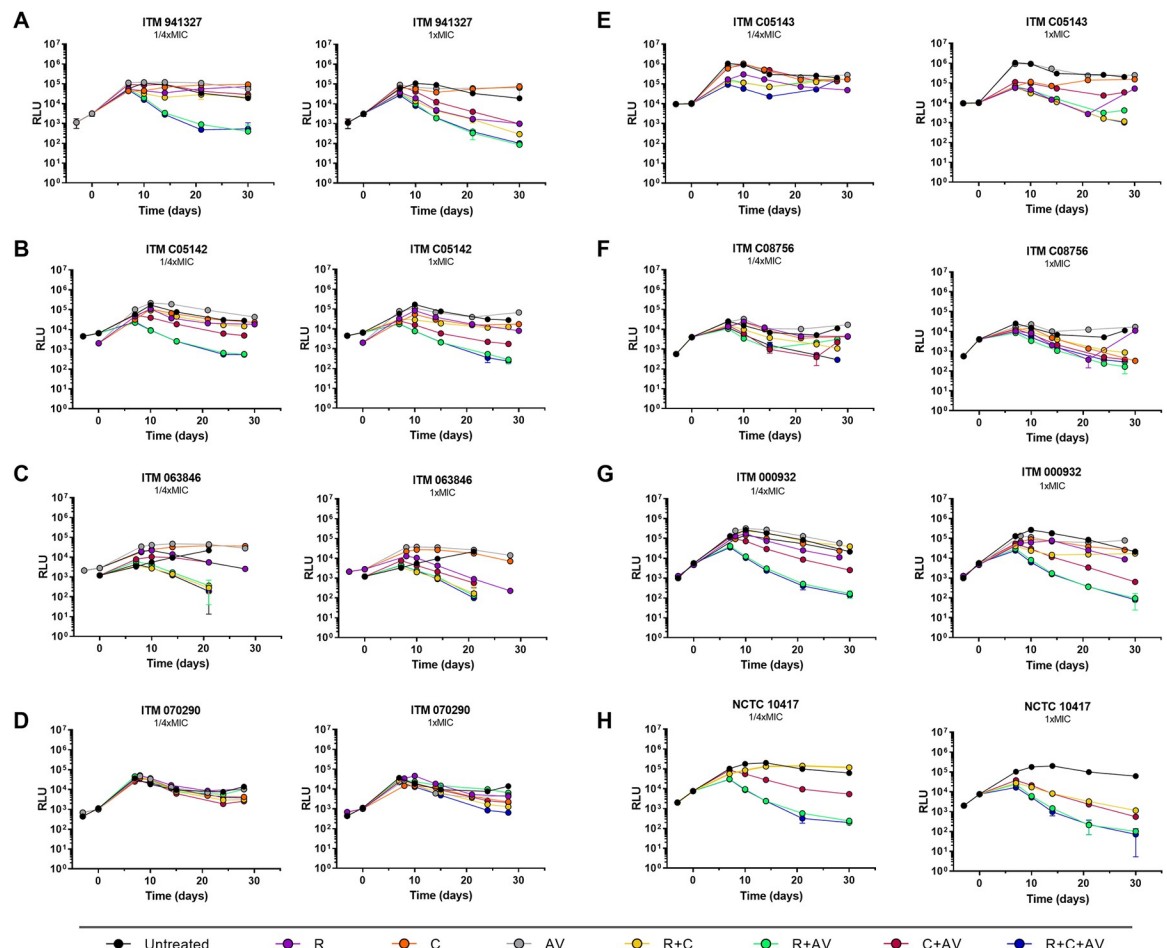

**Fig 2. Time-kill kinetics of rifampicin, clarithromycin and amoxicillin/clavulanate alone and in combination against eight different *M. ulcerans* clinical isolates.** Time-kill kinetics for **(A)** ITM 941327, **(B)** ITM C05142, **(C)** ITM 063846, **(D)** ITM 070290, **(E)** ITM C05143, **(F)** ITM C08756, **(G)** ITM 000932, and **(H)** NCTC 10417. Rifampicin, clarithromycin and amoxicillin were added at 1/4xMIC and 1xMIC. Clavulanate was used at fixed 5 µg/mL. $MIC_{RIF}$ = 0.1 µg/mL; $MIC_{CLA}$ = 0.25 µg/mL; $MIC_{AMX/CLV}$ = 0.5 µg/mL. RLU, Relative Light Units; R, rifampicin; C, clarithromycin; AV, amoxicillin/clavulanate.

A generalised trend of interaction was identified for the pair-wise RIF+AMX/CLV and the triple RIF+CLA+AMX/CLV combinations, which were highly bactericidal, whereas single drugs alone (RIF, CLA and AMX/CLV) had no effect at those concentrations. In fact, this was remarkable when drugs were used at 1/4xMIC values, which correlated with sub-MIC values (1/32 to 1/2 MIC reduction in some cases). At these concentrations, RIF+CLA combination, which is the current treatment for BU, was only active against the ITM 063846 isolate (**Fig 2C**). Interestingly, even the highest RIF concentration tested (100xMIC, MIC increase of 200-fold) against the ITM C05143 isolate was not able to prevent bacterial re-growth, which was not observed when treated at 1xMIC in combination therapy with either of the RIF-containing combinations (**Figs 2E and S1E**). Regarding the ITM 070290 isolate, a RIF concentration of 1 µg/mL (10xMIC, MIC increase of 2-fold) was bactericidal (**S1 Fig**). However, a clear trend of interaction could not be identified, which might be due to effective concentrations of the antibiotics being much lower than their actual MIC values, i.e., RIF was tested at concentrations 5- and 20-fold lower than their actual MIC values, while this being 4- and 16-fold reduction for CLA tested at 1xMIC and 1/4xMIC values, respectively. In the case of AMX/

CLV, no antibacterial activity was observed for this isolate even at the highest concentration, which correlated with a MIC increase of 800-fold.

## Discussion

BU treatment has dramatically improved since 2004 (when only surgery was available) with the introduction of antibiotic therapy and, since 2017, with the current combination therapy of eight weeks of RIF and CLA with a cure rate of almost 96% [17,18]. Nevertheless, adherence to the current 8-week antibiotic course is still a challenge, and shorter and more effective treatments are needed since BU, as many other NTDs, usually affect poor rural communities where access to medicines and medical treatment is difficult [19–21]. Arenaz-Callao *et al.* [3] described the strong *in vitro* synergistic interaction between AMX/CLV and RIF plus CLA against *M. ulcerans*. Its potential as a novel treatment shortening BU therapy was the basis rationale for an ongoing clinical trial in West Africa, the BLMs4BU trial (*ClinicalTrials.gov Identifier*: NCT05169554; *Pan Africa Clinical trial registry*: PACTR202209521256638) [22]. These *in vitro* studies used the checkerboard assay to quantify the degree of drug interaction. However, checkerboard assays rely on measurements of the MIC, a growth inhibition readout, at a single time-point and cannot inform on the antibacterial activity of the drugs alone or in combination (i.e., bacteriostatic or bactericidal) [23]. CFU-based TKA are thus a better proxy to quantify the degree of activity of the interactions. Nonetheless, performing TKA with *M. ulcerans* is challenging since it is a BSL-3 pathogen in many countries and it can take up to three months to form a single colony. Therefore, in this study, different methodologies were compared against the gold standard CFU-based method; all of those that were able to differentiate viability provided comparable results, which depicted the availability of a diverse range of biomarkers in BU research. Moreover, it was confirmed that (i) AMX/CLV enhanced the antibacterial activity of RIF and CLA against *M. ulcerans* [3], (ii) RIF+AMX/CLV-containing combinations were bactericidal and, (iii) that they were more effective *in vitro* than the currently used combination to treat BU of RIF and CLA.

Biomarkers play an essential role in diagnostics and drug development [24,25]. To quantify the bacterial load upon antibiotic treatment, we used CFU enumeration, RLU and $OD_{600}$ measurements, and quantification of 16S rRNA/IS*2404* genes. One of the aims of this work was to explore how well these methodologies agreed with each other. The close correlation between CFU enumeration and the other methodologies showed that all but IS*2404* qPCR were reliable surrogate markers for detecting *M. ulcerans* live cells. Thus, the particular features of the methodologies may drive the choice of the most suitable one, depending on the purpose and the context of the study.

IS*2404* qPCR is the gold standard for the diagnosis of BU in reference laboratories worldwide [26]. As it was expected, the copy number of the IS*2404* gene did not decrease in any condition, which indicated that this methodology is suitable for diagnostic purposes but not for dynamic evaluation of bacterial viability. In our *in vitro* studies, we used a close compartment in which nutrients were not replenished and bacterial waste was not removed. Consequently, it was not possible to discern the genetic material from bacterial cells dead or alive.

The 16S rRNA RT/IS*2404* qPCR assay is a biomarker used to monitor treatment response for patients in a clinical environment [14,15]. Sarpong-Duah *et al.* [15] showed that BU patients samples rapidly became *M. ulcerans* negative during antibiotic treatment with eight weeks of RIF and streptomycin. Despite the restraints including the "delicacy" of RNA, the requirement of a well-equipped laboratory, low throughput as a screening assay and its cost that may limit the applicability and pragmatic implementation in endemic settings as a diagnostic tool, this methodology can be implemented in basic research and clinical laboratories to

monitor bacterial viability over time. The turnaround time (TAT) for the 16S rRNA RT/ IS*2404* qPCR assay is contingent on the source material, being faster when processing clinical samples than liquid cultures. De facto, this methodology is the most suitable to be used with clinical samples. In a laboratory setting, the TAT for either methodology, $OD_{600}$, RLU or 16S rRNA RT/IS*2404* qPCR, would be much shorter, with immediate results compared to the three months needed for the gold-standard methodology (CFU enumeration). More specifically, $OD_{600}$ and RLU measurements require less manipulation and time than 16S rRNA RT/ IS*2404* qPCR. $OD_{600}$ measurement is a fast and simple method but true bacterial quantification is not possible and counts should be verified by alternative methods [27]. In fact, it did not show a good correlation with 16S rRNA copy numbers and RLU. Nevertheless, growth trends were still discernible, and $OD_{600}$ values still pointed out that cell proliferation stopped sooner in the combinations including AMX/CLV compared to other conditions. Luminescence (RLU) assays are costly and interlaboratory comparisons may be difficult due to the luminometer set up. However, ATP production is an indirect measurement of cell viability [7,28] and, indeed, bioluminescence as an ATP-dependent process has been used to evaluate drug efficacy in several *M. ulcerans* studies [9–12,29–31]. Our results showed that RLU correlated well with CFU enumeration, and also with the number of 16S rRNA copies, which had the highest correlation coefficients (r $\geq \sim 0.700$) in all conditions, both being biomarkers of active metabolism. In fact, based on cost-effective considerations including time to results, sample processing and manipulation time, this was the methodology selected to expand the study to further clinical isolates.

RIF+AMX/CLV and the triple RIF+CLA+AMX/CLV combinations showed the highest antibacterial activity against all the isolates. When testing the compounds alone, concentrations of RIF and CLA correlating with a MIC increase of 2-16-fold were needed to observe cell death, whereas AMX/CLV was hardly active by itself. This antibiotic still displayed a strong synergistic interaction with RIF and RIF/CLA at concentrations closed to MIC or even sub-MIC values, suggesting that the sterilization activity of RIF could be increased in combination with AMX/CLV, due to their synergistic interaction, while maintaining its dose [3]. In the same line, but with a different strategy, Omansen *et al.* [31] showed that higher doses of RIF than currently recommended were efficacious in reducing BU disease in a mouse model.

In general, specific responses to drugs alone and in combination for each isolate were variable in our assays. This is not unexpected when considering the intrinsic variability in MIC distributions among *M. ulcerans* clinical isolates [32]. In the case of these bacteria, there are not established epidemiological cut-off (ECOFF) values and clinical breakpoints according to international antimicrobial susceptibility testing guidelines [33,34]. However, antibiotic concentrations used in this study were much lower than the ones reported in plasma in several pharmacokinetic studies [35–39]. Furthermore, this variability does not seem to play a significant role when translating to clinical efficacy considering that the efficacy of the standard treatment for BU (RIF+CLA for eight weeks) is as high as 96% [18].

The main limitation of our study was that TKA, likewise MIC and checkerboard assays, are *in vitro* models where drugs are typically added only once at a certain time point (normally at the beginning of the experiment at time zero). It is well-known that antibiotics can degrade in the assay media. For slow growing bacteria such as *M. ulcerans*, timescales are larger than a typical bacterial growth assay [40]. RIF has a half-life of seven days and $\geq$75% is degraded after 14 days of incubation in 7H9 medium at 37°C [41–43]. A similar degradation time or even shorter (14.2 h) was described for CLA [41,43]. Different degradation half-lives (from 27.4 h to 10 days) have been reported for AMX based on different media conditions [44–47]. These reported degradation kinetics indicate that, in our study, the remaining drug concentrations might be negligible at the end of the 28 days of the assay. Nevertheless, the maximum effect of

the combinations was observed after 28 days, which indicated a strong post-antibiotic effect in the absence of effective drug concentrations, an important feature when translating into clinical application where drug exposure is limited by the dose and posology of the medicine.

In summary, we have performed a head-to-head comparison of different methodologies to quantify *M. ulcerans* bacterial load in *in vitro* cultures against the gold-standard CFU enumeration methodology. We have demonstrated that *M. ulcerans* TKA can be performed using a wide variety of biomarkers for BU research, which use will depend on the context of the investigation and the setting to be implemented. Our TKA confirmed the strong *in vitro* synergistic interaction of AMX/CLV in combination with either RIF or RIF/CLA against *M. ulcerans*, and further support current clinical activities to shorten BU treatment [22].

## Supporting information

**S1 Table. Correlation between Minimum Inhibitory Concentration (MIC) values and concentrations of rifampicin, clarithromycin, and amoxicillin used in the time-kill kinetics at 1/20xMIC, 1/4xMIC, 1xMIC, 4xMIC, 10xMIC, 20xMIC, and 100xMIC.** Drug concentrations used in the experiments and the corresponding ones are described in parentheses (µg/mL). Clavulanate was used at fixed 5 µg/mL. The synergistic MIC (MICsyn) was defined as the Most Optical Combinatorial Concentration (MOOC), i.e., the lowest possible concentration of every compound that, when assayed together, prevented bacterial growth [3]. aIn the presence of 5 µg/mL of clavulanate MIC, Minimum Inhibitory Concentration; MICsyn, Synergistic Minimum Inhibitory Concentration; MOCC, Most Optimal Combinatorial Concentration; RIF, rifampicin; CLA, clarithromycin; AMX, amoxicillin; NA, not available.
(XLSX)

**S2 Table. $Log_{10}$ CFU/mL reduction of *M. ulcerans* isolate ITM 000932 bacterial burden upon treatment with rifampicin, clarithromycin and amoxicillin/clavulanate.** Values represent changes between day 28 and days 0, 1, 3, 7 and 10. [a]$p < 0.05$. CFU, Colony Forming Units; RIF, rifampicin; CLA, clarithromycin; AMX/CLV, amoxicillin/clavulanate.
(XLSX)

**S1 Fig. Related to Fig 2. Time-kill kinetics of different *M. ulcerans* isolates upon treatment with different concentrations of rifampicin, clarithromycin and amoxicillin/clavulanate alone.** (A) ITM 941327, (B) ITM C05142, (C) ITM 063846, (D) ITM 070290, (E) ITM C05143, (F) ITM C08756, (G) ITM 000932. RLU, Relative Light Units; R, rifampicin; C, clarithromycin; AV, amoxicillin/clavulanate.
(TIF)

## Acknowledgments

We would like to thank Begoña Gracia Díaz for her technical support during the initial stages of the work. Authors would like to acknowledge the use of Servicio General de Apoyo a la Investigación-SAI, Universidad de Zaragoza.

## Author Contributions

**Conceptualization:** Santiago Ramón-García.

**Data curation:** Emma Sáez-López, Ana C. Millán-Placer, Ainhoa Lucía.

**Formal analysis:** Emma Sáez-López, Santiago Ramón-García.

**Funding acquisition:** Santiago Ramón-García.

**Investigation:** Emma Sáez-López, Ana C. Millán-Placer, Ainhoa Lucía.

**Methodology:** Emma Sáez-López.

**Project administration:** Emma Sáez-López, Santiago Ramón-García.

**Supervision:** Santiago Ramón-García.

**Visualization:** Emma Sáez-López.

**Writing – original draft:** Emma Sáez-López, Santiago Ramón-García.

**Writing – review & editing:** Emma Sáez-López, Ana C. Millán-Placer, Ainhoa Lucía, Santiago Ramón-García.

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
