## [Decision Letter · Decision Letter 0]

30 Jan 2024

Dear Dr Saez Lopez,

Thank you very much for submitting your manuscript "Amoxicillin/clavulanate in combination with rifampicin/clarithromycin is bactericidal against Mycobacterium ulcerans" for consideration at PLOS Neglected Tropical Diseases. As with all papers reviewed by the journal, your manuscript was reviewed by members of the editorial board and by several independent reviewers. The reviewers appreciated the attention to an important topic. Based on the reviews, we are likely to accept this manuscript for publication, providing that you modify the manuscript according to the review recommendations. 

Thank you for your submission which has been reviewed by three experts in the field. They all found the work to be a valuable contribution but have specific questions that need to be addressed as well as clarifications to be made.

We look forward to your careful and clear responses to all the concerns.

Sincerely,

Paul J. Converse

Academic Editor

Stuart Blacksell

Section Editor

Dear Dr. Saez Lopez,

Thank you for your submission which has been reviewed by three experts in the field. They all found the work to be a valuable contribution but have specific questions that need to be addressed as well as clarifications to be made.

We look forward to your careful and clear responses to all the concerns.

Reviewer's Responses to Questions

**Key Review Criteria Required for Acceptance?**

**Methods**

-Are the objectives of the study clearly articulated with a clear testable hypothesis stated?

-Is the study design appropriate to address the stated objectives?

-Is the population clearly described and appropriate for the hypothesis being tested?

-Is the sample size sufficient to ensure adequate power to address the hypothesis being tested?

-Were correct statistical analysis used to support conclusions?

-Are there concerns about ethical or regulatory requirements being met?

Reviewer #1: The objectives are clear and the methods are well both described and suitable for the purposes of the study

Reviewer #2: Based on MIC determination and checkerboard assays it has been suggested to include amoxicillin/clavulanate (AMX/CLV) in the recommended BU chemotherapy with rifampicin and clarithromycin. These data served as in vitro microbiological support for the clinical testing of the triple RIF+CLA+AMX/CLV combination treatment.

In the present report, the authors have now compared different time-kill assay readouts (CFU enumeration, luminescence relative light unit determination based on ATP quantification, optical density measurements, IS2404 qPCR and 16S rRNA RT-qPCR) to capture the effect of AMX/CLV. The RIF+AMX/CLV and RIF+CLA+AMX/CLV combinations showed the strongest bactericidal activity against a standard strain (ITM 000932) with a CFU reduction at Day 28 of about three Log10. Based on the data obtained with isolate 000932, the authors used the luminiscence assay to analyze the effect of the RIF+CLA+AMX/CLV combination on a set of M. ulcerans clinical isolates at 1/4xMIC and 1xMIC concentrations of the compounds. 

The objectives of the study clearly articulated, and the study design is appropriate.

Reviewer #3: This is a technical laboratory study that sets out to evaluate different methods for in vitro drug susceptibility estimation with a particular eye on the difficulties of doing this with slowly growing bacteria such as M. ulcerans. The study design is reasonable in this regard.

**Results**

-Does the analysis presented match the analysis plan?

-Are the results clearly and completely presented?

-Are the figures (Tables, Images) of sufficient quality for clarity?

Reviewer #1: The figures are clear and well presented but the results are sometimes difficult to understand because the authors use set concentrations which they refer to as the MIC, but these rarely correspond to the actual MIC of the different isolates. Thus, for example, in Table S2, the actual MICs are compared to the MIC as used in the assay, but both are referred to as just MIC in the headings. This is discussed at length in the text but is still confusing to read.

Reviewer #2: While the RIF+AMX/CLV and RIF+CLA+AMX/CLV combinations showed also the strongest effect in luminescence, optical density 16S rRNA RT-qPCR assay, no correlation was observed with the IS2404 qPCR readout. The statements ‘In summary, an overall good correlation was observed between CFU counting and the other methodologies (line 119 and 120), ‘in fact, all provided comparable results’ (line 283), and ‘close correlation between CFU counting and the other methodologies’ (line 292) should be modified accordingly. In this context, it should also be discussed more clearly, from which readouts one expects to see a decline associated with the killing of the mycobacteria.

Time-kill kinetic data confirmed the high activity of RIF+AMX/CLV and RIF+CLA+AMX/CLV combinations, but the authors should also mention that RIF+CLA yielded comparable results for two isolates (CO5143 and 063846). The exceptional properties of strain 070290 should be commented and it should be made clearer, why isolate CO5143 is also labelled ‘exceptional’ (line 251).

Minor point: in line 252, CO15143 should read CO5143.

Figures are of sufficient quality, but minor modifications are required in the description of the results.

Reviewer #3: yes there is reasonable evidence to support the main claims made in this manuscript.

There is a sentence in the results that didn't make sense to me and needs clarification.

Lines 248 and 249

A generalised trend of interaction was identified for pair-wise RIF+AMX/CLV and triple

RIF+CLA+AMX/CLV combinations, which were highly bactericidal, whereas drugs

 alone had no effect at those concentrations

Do the authors mean SINGLE drugs alone? If so which specific single drugs? Is this an argument to show that combinations are more potent than single drugs?

**Conclusions**

-Are the conclusions supported by the data presented?

-Are the limitations of analysis clearly described?

-Do the authors discuss how these data can be helpful to advance our understanding of the topic under study?

-Is public health relevance addressed?

Reviewer #1: The conclusions are broadly supported and the limitations acknowledged. The potential usefulness of the luminescence assay in a clinical setting may be a little exaggerated as although it has advantages over the other methods, it still requires Biosafety level 3 facilities and specialized equipment and reagents that may not always be readily available. It is worrying that the responses of the different strains to the drug combinations used are quite variable. The impact this may have on clinical use should be discussed in detail.

Reviewer #2: (No Response)

Reviewer #3: the conclusions are supported by the data presented. I would like to see some more information about laboratory turn around time (potentially) if the faster methods described in this research were to be implemented.

**Editorial and Data Presentation Modifications?**

Reviewer #1: It would be very helpful if the assay MICs and the actual MICs could be distinguished more clearly in the text and figures

Reviewer #2: (No Response)

Reviewer #3: (No Response)

**Summary and General Comments**

Reviewer #1: The combined use of amoxicillin/clavulanate with rifampacin/clarithromycin for the treatment of Buruli Ulcer is currently undergoing clinical trials. This paper is an in vitro study that compares several different methodologies for following the effect of the drugs individually and in combination on M.ulcerans concentrations. The authors report that a luminescence based assay is the most cost-effective method that correlates well with the gold standard CFU counting and go on to compare the results obtained with this assay using a panel of bacterial strains. The work is well done and adds an alternative method to those currently routinely used to monitor M. ulcerans growth. The writing is generally good but sometimes lacks clarity.

Reviewer #2: Overall, the manuscript provides valuable information for the choice of the different types of measurement for time-kill assays, dependent on the purpose of a study. In particular, the discussion on the degradation kinetics of antibiotics may animate discussions on study design.

Reviewer #3: This is a technical study that is potentially useful. My skill set is predominantly clinical infectious diseases including the management of Buruli patients, hence I suggest also a review by a microbiology scientist or medical microbiologist. Nevertheless it is interesting and convincing to see this in vitro demonstration of the effectiveness at a microbiological level of the combinations of rifampicin plus amox/clav. I think it is unlikely that we will ever use triple drug therapy in clinical practice as rifampicin and clarithromycin are already highly effective. However there are times we need to replace one or other of these drugs for patients who develop antibiotic intolerance. It is good to know that there is consistent laboratory support at least for Rifampicin with another safe and licensed antibiotic.

PLOS authors have the option to publish the peer review history of their article (what does this mean?). If published, this will include your full peer review and any attached files.

Reviewer #1: No

Reviewer #2: No

Reviewer #3: No

Figure Files:

Data Requirements:

Reproducibility:

References

---

## [Editor Report · Decision Letter 1]

7 Mar 2024

Dear Dr Saez Lopez,

We are pleased to inform you that your manuscript 'Amoxicillin/clavulanate in combination with rifampicin/clarithromycin is bactericidal against Mycobacterium ulcerans' has been provisionally accepted for publication in PLOS Neglected Tropical Diseases.

Best regards,

Paul J. Converse

Academic Editor

Stuart Blacksell

Section Editor

Thank you for carefully addressing all of the reviewers' comments.

---

## [Editor Report · Acceptance letter]

21 Mar 2024

Dear Dr Saez Lopez,

We are delighted to inform you that your manuscript, "Amoxicillin/clavulanate in combination with rifampicin/clarithromycin is bactericidal against *Mycobacterium ulcerans* ," has been formally accepted for publication in PLOS Neglected Tropical Diseases.

Best regards,

Shaden Kamhawi

co-Editor-in-Chief

Paul Brindley

co-Editor-in-Chief
